# Ultrasonic Extraction of 2-Acetyl-1-Pyrroline (2AP) from *Pandanus amaryllifolius Roxb.* Using Ethanol as Solvent

**DOI:** 10.3390/molecules27154906

**Published:** 2022-07-31

**Authors:** Aisyah Nur Hanis Azhar, Nurul Aini Amran, Suzana Yusup, Mohd Hizami Mohd Yusoff

**Affiliations:** 1Chemical Engineering Department, Universiti Teknologi PETRONAS, Seri Iskandar 32610, Perak, Malaysia; aisyah_19001058@utp.edu.my (A.N.H.A.); drsuzana_yusuf@utp.edu.my (S.Y.); hizami.yusoff@utp.edu.my (M.H.M.Y.); 2HICOE—Center for Biofuel and Biochemical Research, Institute of Self-Sustainable Building, Universiti Teknologi PETRONAS, Seri Iskandar 32610, Perak, Malaysia

**Keywords:** ultrasonic extraction, pandan leaves, 2-Acetyl-1-Pyrroline (2AP), solvent, RSM

## Abstract

2-acetyl-1-pyrroline (2AP) is the compound that gives out the typical aroma and flavour of pandan leaves (*Pandanus amaryllifolius Roxb*.). This research incorporates ultrasonic extraction to extract the aromatic compound in pandan leaves. The parameters varied in this study are the extraction time, sonicator amplitude, concentration of solvent and the mass of pandan leaves. The experiment was conducted using a central composite design (CCD) model generated by the response surface methodology (RSM). From the extraction process, it can be deduced that the effect of leaves’ mass is comparably higher than other parameters, while sonicator amplitude gives the most negligible impact on the process. The obtained *p*-value was 0.0014, which was less than 0.05. The high R-squared 0.9603 and adjusted R-squared 0.8809 indicate the model is well agreed with the actual data. The optimal control variables of ultrasonic extraction of 2AP are at an extraction time of 20 min, 60% of solvent concentration, amplitude of 25% and 12.5 g of pandan leaves, which produced 60.51% of yield of the extract and 1.43 ppm of 2AP. It is found that the mass of pandan leaves and the concentration of solvent have a significant impact on the extraction process of 2AP.

## 1. Introduction

Pandan leaves belong to the *Pandanus* (*Pandanaceae* family), palm-like evergreen trees or shrubs. They are widely distributed from Africa to the Pacific lands in the moist tropics. There are over 600–700 species, and the sizes vary from less than 1 m tall to 20 m tall. The leaves are strap-shaped (elongated, narrowing towards the tip), varying from 30 cm long to 2 m or longer and 1.5 cm up to 10 cm broad, depending on the species [1]. Of all the species, the only Pandanus species with fragrant leaves is *Pandanus amaryllifolius Roxb.* It holds commercial interest in the flavour industry. This species is abundant in Southeast Asia, such as Malaysia, Thailand, Indonesia, and India [2]. In Asian countries, pandan leaves are often used as a food colourant. For instance, the leaves are applied to food as fresh whole or juice and have become popular as a green food colourant and pandan flavour in cakes, ice cream and custards, and other Asian cuisines. The leaves are also well known among folk medicine for their healing properties. The water extract from fresh pandan leaves can treat internal inflammation, coughs, colds and measles [3], while the juice is combined with aloe vera to cure some skin diseases. Additionally, fresh leaves are believed to work as insect repellent for certain types of household insects.

Pandan leaves are of a few volatile compounds in a group of alcohols, aromatics, carboxylic acids, ketones, aldehydes, esters, hydrocarbons, furans, furanone and terpenoids. Other volatile compounds in pandan extracts include ethyl formate, 3-hexanol, 4-methylpentanol, 3-hexanone, 2-hexanone, trans-2-heptenal, β-damascenone, 4-ethylguaiacol and 3-methyl-2-(5H)-furanone [3,4]. The primary compound that contributes to pandan leaves’ flavour characteristics is 2-acetyl-1-pyrroline (2AP). It is a volatile compound with desirable aroma terms such as “pandan-like” for Orientals or “popcorn-like” for the non-Orientals. 2AP is the aroma compound and the flavour that gives out the typical smell from white bread, jasmine rice, basmati rice, pandan and bread flowers. As for Pandan leaves, the leaves only produce the aroma once they are withered, cooked or pounded [1].

In this study, despite many conventional extraction techniques available, a greener technique, ultrasonic-assisted extraction (UAE), is favoured to extract 2AP from pandan leaves. The benefit of the UAE is mainly due to a reduction in extraction time and solvent consumption compared to conventional methods. The UAE working principle depends on acoustic cavitation that happens when significant negative pressure is applied, forming microbubbles in liquid. The bubbles will grow to various sizes, collapse violently, and release intense local energy with significant chemical and mechanical effects. Temperature and pressure changes from these implosions cause shear disruption, thinning of cell membranes, and cell disruption, resulting in enhanced solvent penetration into cells and amplifying the mass transfer of target compounds into the solvent. The direct outcome of these processes is that the mass transfer from plant material to bulk solvent is significantly improved, and the extraction yield is raised [5,6,7,8].

The solvent to be used in this process is a mixture of ethanol and water. Water is known to be abundant and its properties are non-toxic and polar and will not bring harm to the environment. The compound to be extracted in this study is 2AP, which is a polar compound; therefore, 2AP should be able to dissolve in water. Ethanol is also a polar solvent; therefore, it can dissolve 2AP. The polarity of water and ethanol are higher compared to n-hexane which makes it a better solvent to extract 2AP [2,9].

Despite the advantages of the process, there has been no available literature on the use of UAE to extract 2AP from pandan leaves. Literature on the extraction of 2AP from pandan leaves has been focused on supercritical carbon dioxide and solvent extraction methods. Ultrasonic extraction has never been reported as one of the methods to extract 2AP. Hence, optimization of the extraction parameters needs to be carried out to obtain the maximum yield of 2AP from pandan extract. The parameters involved in this study include sonication amplitude, extraction time, solid to solvent ratio and concentration of solvent. Therefore, this study aimed to find the optimum conditions of ultrasonic extraction to obtain a maximum concentration of 2AP from pandan extract. This method may allow for the substitution of natural pandan flavours for artificial pandan flavours, which may serve to raise the value of various low-quality food products, such as old and non-aromatic rice, as well as contribute value to the plant as the source of the pandan flavour.

## 2. Materials and Methods

### 2.1. Preparation of Pandan Leaves

Pandan leaves (*Pandanus amaryllifolius Roxb.*) were collected randomly from their trees grown in Perak, Malaysia on 4 October 2020. The fresh green leaves were then weighed using an electronic balance. The leaves were put on a tray, and the tray was put into an oven at a temperature of 105 °C for 24 h. After 24 h, the tray containing the leaves was taken out of the oven, and the leaves’ weight was measured using an electronic balance [9]. The moisture content was calculated. Dried pandan leaves were cut into smaller pieces before being ground into desired particle size using a grinder. The desired particle size of the leaves was around 0.5–1.0 mm^2^ [10]. After the grinding process was finished, the particles were sieved into desired particle size. The leaves were then put in a sealed bag and kept at room temperature until used.

### 2.2. Chemical Reagants

The solvent used for ultrasonic extraction was ethanol with 96% purity. Ethanol was purchased from Fisher Scientific. For qualitative and quantitative analysis, acetone and 2,4,6-trimethylpyridine (TMP) were used. Both reagents were purchased from Sigma Aldrich. The purity was 99%. All the solvents and chemicals used in this study were analytical grade.

### 2.3. Experimental Design

In this study, four parameters were manipulated to observe their effect on yield extract and the concentration of 2AP obtained from the extract. The parameters involved were extraction time, the concentration of solvent, sonication amplitude and mass of leaves. This study used an ultrasonic probe because it allowed direct contact with the sample and allowed it to develop a power up to 100 times more than that provided in the ultrasonic bath. After the extraction process was done, the solution was filtered using vacuum pump and the solvent was removed using a rotary evaporator. Then, the yield of the pandan extract was then calculated before the extract was analysed using GC-MS and GC-FID.

### 2.4. Ultrasonic Extractions of 2AP from Pandan Leaves

Dried pandan powder was weighed according to the desired mass using an electronic balance and stored in a 500 mL ultrasonic vessel. The solvent used in this process was ethanol and distilled water (*v*/*v*)%. 200 mL of solvent was poured into the vessel. The vessel was then connected to a refrigerated water bath at a temperature of 35 °C to help to maintain the process temperature. The setup for the ultrasonic system is shown in Figure 1 below, while the specification is shown in Table 1 below.

Cole-Parmer ultrasonic processor was switched on. The temperature was set at 45 °C since the temperature range for extraction of bioactive compounds was usually within 40 until 80 °C. The amplitude was then set to the desired value. The probe was put into the vessel and turned on. After the designated extraction time finished, the processor and the refrigerated water bath were turned off.

The extract was then filtered using a filter pump under reduced pressure. After that, the solvent was evaporated using a Buchi rotary evaporator under reduced pressure at 60 °C until dryness. The concentrated extracts were weighed using an electronic balance. The yield of the extract was calculated.

### 2.5. Qualitative and Quantitative Analysis

#### 2.5.1. Analytical Method

The yield of pandan extract was calculated using the following formula:Yield (%) = M_1_/M_2_ × 100%(1)

Whereby M_1_ is the mass of pandan extract from the sample (g) and M_2_ is the mass of the pandan leaves.

#### 2.5.2. GC-MS Analysis

Gas chromatography–mass spectrometry (GC-MS) was used to analyse the compounds present in the extract from pandan leaves. The operating parameters involved in doing GS-MS analysis were as follows: injector temperature is 250 °C, carrier gas used was helium at a flowrate of 1.0 mL/min operated in the split mode 1:5, the mass spectrometry parameters were electron impact of 70 eV, ion source temperature of 200 °C, line of transference is 250 °C, electron multiplier voltage of 2500 V, and the scan rate of 5 scans/s and mass interval of 35–500 m/s [10]. After the GC-MS analysis is done, the peak present in the result will indicate the compounds present in the extract.

#### 2.5.3. GC-FID Analysis

A method used to minimize the errors when an accurate weight or volume of samples is required to determine the amount of 2AP present in the extract [11,12,13], which in this study is by using 2,4,6-trimethylpyridine (TMP) in the quantitative analysis of 2AP by GC-FID. A stock solution of 250 ppm TMP is prepared by adding 10 μL of TMP to 40 mL of acetone. This stock solution will be diluted to 50 ppm before adding it to each 0.1 mL of the extracts, and then 3 μL of the samples will be injected into GC-FID. The following formula can calculate the amount of 2AP present in the sample:(2)Amount of 2AP (μL/mL)=Concentration of TMP(μLmL)TMP GC Area×2AP GC AreaInjection volume of sample (mL)

### 2.6. Optimization of UAE Extraction Conditions Using Response Surface Methodology (RSM)

#### Central Composite Design (CCD) Response Surface Design

Ultrasonic extraction of 2AP from pandan leaves was carried out using RSM coupled with CCD, which is a particular set of mathematical and statistical methods for designing experiments, building models, evaluating the effects of variables, and searching for optimum results conditions of variables to predict targeted response. A CCD with four factors was performed to investigate and confirm the process parameters affecting the extraction of pandan extract. Four independent variables: extraction time, the concentration of solvent, mass of pandan leaves, and sonication amplitude were studied with the yield of pandan extract as the response. Optimum conditions for extracting pandan extract were then determined. Design-Expert 12.0, a statistical software, was used for regression analysis and graphical analysis of the data obtained during the ultrasonic extraction experiment. The statistical significance of the full quadratic model predicated was evaluated by the analysis of variance (ANOVA) and least-squares techniques. The ANOVA lies in determining which factors significantly affect the response variables in the study.

## 3. Results and Discussion

### 3.1. Moisture Content of Pandan Leaves

The moisture content of the pandan leaves was carried out to determine the effect of the drying pre-treatment on the yield and concentration of 2AP in pandan leaves extract. The moisture content of fresh pandan leaves was calculated to be 81.4% after the procedure of oven drying at 105 °C for 24 h was done. The result was typical for plant materials where about 80% of water is removed. The previous study included the moisture content of pandan leaves at 86.5% [10].

### 3.2. Data Summary of Optimization Process

In this study, four factors were used to evaluate the effect of process variables on the response, namely the yield of pandan extract. A total of 26 batch experiments were carried out, as shown in Table 2 below. Then CCD experimental data were analysed by multi-regression analysis.

### 3.3. Statistical Analysis

Table 3 below summarizes the data obtained after ultrasonic extraction of pandan leaves associated with response variables: yield of extract and concentration of 2AP in pandan extract. The control variables are extraction time (A), the concentration of solvent (B), sonicator amplitude (C) and mass of pandan leaves (D). Analysis of variance (ANOVA) was carried out with constraints set as *p*-value < 0.05 for models to be significant and *p*-value > 0.05 for insignificant lack of value. The regression analysis of the responses was carried out using a second-order polynomial equation for the evaluation of the best fit model: (3)$$Y_{n}=\beta_{0}+\sum_{i=1}^{k}\beta_{i}X_{i}+\sum_{i=1}^{k}\beta_{ii}X_{i}^{2}+\sum_{i=1}^{k}\sum_{j=i+1}^{k-1}\beta_{ij}X_{i}X_{j}$$

#### 3.3.1. Model Fitting

A second-order polynomial equation (modified) was developed to understand the interactive correlation between the yield of pandan extract and the process variables. The final model obtained in terms of coded factors is given below,
Y_1_ (%) = 61 − 1.71A + 6.14B − 0.5454C − 9.52D + 3.47AC + 0.7514AD − 5.53BC + 2.09CD − 2.11A^2^ − 2.26B^2^ + 0.2523C^2^ − 3.55D^2^(4)

A second-order polynomial equation (quadratic) was developed to understand the interactive correlation between concentration of 2AP and the process variables. The final model obtained in terms of coded factors is given below,
Y_2_(ppm) = 2.07 − 0.5724A + 0.4048B + 0.1174C − 0.7563D − 0.0951AC + 1.43AD + 0.2247BC − 1.05BD − 0.1313CD + 0.2852A^2^ − 0.1288B^2^ + 0.2318C^2^ + 0.7213D^2^(5)

Y_1_ is the yield of pandan extract, Y_2_ is the concentration of 2AP, A is extraction time, B is the solvent concentration, C is sonication amplitude, and D is the mass of pandan leaves. 

#### 3.3.2. Optimality of Independent Variables

Among all the variables studied for ultrasonic extraction, it can be affirmed that the mass of dried pandan leaves has the most significant impact on the extraction process. The solvent concentration plays an essential role since higher concentration favoured better extraction, as there is higher availability of such compounds with the higher loading of the solute. Another control variable that influences the measured responses is the extraction time. However, operation time is critical during the process as it would significantly reduce electricity consumption due to reduced operation time [14].

The optimization procedure was performed and the optimum conditions of the ultrasonic extraction process of pandan extract yield from pandan leaves were found to be as mentioned in Table 4 below. The extraction time was 20 min with 60% of solvent concentration, amplitude at 25% and 12.5 g of pandan leaves.

The optimum conditions were repeated three times and the average value of pandan extract yield was found to be 60.51%, as mentioned in Table 5 below.

From this process, it can be concluded that the experimental results were in good agreement with the predicted value. The optimum extraction time was 20 min. A study of the extraction of carotenoids [15] mentioned that the optimum ultrasonic extraction time was 20 min, which produced 334 mg/L of β-carotene. Depending on the plant matrix, a long extraction time is required as it enhances extraction yields, which is favourable in the process. Long extraction times, on the other hand, may cause undesirable changes in the extracted compound [16].

The optimum solvent concentration is in agreement with previous research conducted. According to a study in ultrasonic extraction of polyphenols from apple pomace [17], the highest extraction yield of polyphenol was found at 60% ethanol concentration as solvent. Because of its different molecular structure, ethanol is more effective at eluting compounds like chlorogenic acid and flavonoids [16]. Water is a more effective solvent due to its higher polarity coefficient [18]. As a result, combining water and ethanol yields a higher polyphenol extraction efficiency than using these solvents separately.

Next, the optimization procedure was performed and the optimum conditions of the ultrasonic extraction process of 2-acetyl-1-pyrroline (2AP) from pandan leaves were found to be as mentioned in Table 4 above. The extraction time was 20 min with 60% of solvent concentration, amplitude at 25%, and 12.5 g of pandan leaves.

The optimum conditions were repeated three times, and the average value of the concentration of 2AP was found to be 1.43 ppm, as mentioned in Table 5 above.

### 3.4. Comparison with Other Extraction Methods

From these experimental results, it is found that the optimum conditions of ultrasonic extraction produced the average concentration of 2AP (1.43 ppm) which is higher compared to the previous study using supercritical carbon dioxide extraction (0.16 to 0.19 ppm) [19]. Another study using supercritical carbon dioxide extraction produced 0.72 ppm of 2AP in pandan leaves [20]. Meanwhile, using hexane extraction showed that the highest 2AP concentration found was 0.52 ppm [19]. This shows that ultrasonic extraction is more favourable compared to these two extraction methods as shown in Table 6 below.

Another research has been conducted to extract 2AP from pandan leaves using a solvent extraction process. The solvents used were ethanol, methanol and propanol [2]. It was found that 2AP existed in ethanol and methanol according to the chromatograms obtained from GC-MS analysis, while no 2AP was detected in propanol. However, the exact concentration of 2AP was not specified. It was stated that the yield of 2AP is higher in ethanol compared to methanol.

This is understandable given that the presence peak in the ethanol chromatogram was higher than in the methanol chromatogram. Ethanol is an extremely polar molecule. This is due to ethanol’s hydroxyl (OH) group. Because of the high electronegativity of oxygen in this alcohol, hydrogen bonding with other molecules is possible. As a result, ethanol attracts polar and ionic molecules, making ethanol have the ability to dissolve both polar and non-polar substances. Furthermore, ethanol has been identified as one of the most important solvents during solvent extraction. Ethanol solvent is also more environmentally friendly than pure alcohol [2]. 

### 3.5. Relationship between Responses

The statistical significance of the developed model equations was evaluated using Pareto analysis of variance (ANOVA). The higher model F value and lower *p*-value (*p* < 0.05) demonstrated that the developed mathematical model was highly significant. The model’s goodness of fit was evaluated by the determination coefficient (R^2^), adj-R^2^ and Pre-R^2^. The high R^2^ values revealed that only slight variations are not explained by the developed mathematical model and demonstrate a significant and intense correlation between the observed and predicted values [9].

Figure 2a–e shows the effect of response variables on the yield of pandan extract. As seen in Table 2, the effect of leaves’ mass is comparably higher than other parameters as indicated by the highest F-value, which is 61.61. At the same time, sonicator amplitude gives the most negligible impact on the process with the lowest F-value, which is 0.2102. The associated interaction parameters of A (extraction time) and D (mass of pandan leaves) can be analysed to be the lowest, while the interaction between B (concentration of solvent) and C (sonicator amplitude) is the highest.

Figure 2a shows the surface plot of yield value as a function of the concentration of solvent (%) and mass of pandan leaves (g) at a fixed extraction time of 30 min and amplitude of 30%. As the concentration of solvent increases, the yield of pandan extract also increases. This study uses ethanol as the solvent. Because ethanol has a lower polarity than water, it promotes the solubility and diffusion of bioactive compounds by lowering the solvent’s dielectric constant. However, earlier research has found that employing highly pure organic solvents can cause dehydration and collapse of plant cells as well as denaturation of cell wall proteins, making the extraction of bioactive compounds challenging [15]. As a result of the differing polarity of the phenolic compounds and the suitability of this system for human ingestion, hydro-alcoholic combinations, particularly ethanol, are the solvent systems most suitable for extraction [16]. Water acts as a swelling agent of the plant matrix, increasing the contact surface, while ethanol induces the rupture of the bond between the solutes and the matrix [17]. 

Meanwhile, the yield of pandan extract shows a declining pattern as the mass of pandan leaves increases. The yield of pandan extract declines from 60% to 40% as the mass of pandan leaves increases from 7.5 g to 12.5 g. The amount of sample will determine the number of compounds that can be released into the solvent. In the case of extracting too much of the sample, the solvent can be readily saturated, avoiding the complete compound extraction. In general, a higher volume of solvent can effectively dissolve more target components and leads to higher extraction efficiency. A high solid to solvent ratio could promote an increasing concentration gradient, which causes an increase in diffusion rate and ultimately increases extraction efficiency. The chance of bioactive compounds coming in contact with extracting solvent is higher when the amount of solvent is increased and causes a higher rate of leaching out. However, it should be noted that once equilibrium is reached, the yield of active compounds will no longer increase.

Figure 2b shows the surface plot of the yield value of pandan extract as a function of amplitude and extraction time. As can be seen from the figure, the 3D surface graphic shows a rather flat surface which depicts the least significant interaction between the two process variables. It can be seen that as the extraction time increase, the yield of pandan extract decreases from 64.8% to 54.5%. The first 20 min could be the washing stage during the extraction process, which means that the extraction process could recover up to 90% of the extract during this period. This is also the stage where the dissolution of the soluble components on the surface of the plant matrix takes place [17,18]. The second stage should follow which is called slow extraction which can take up to 60 min; however, it should be noted that the purpose of the process is to limit the sonication duration and also to reduce the energy cost of the process.

As for the amplitude, the yield of pandan extract decreases from 64.8% to 58% as the amplitude increases. According to Table 3, the amplitude is the least significant variable as the *p* value is higher than 0.05. This can be supported by a study to extract total polyphenols and antioxidant activity from mango residues [18]. According to the study, the sonication amplitude was found to be among the parameters that do not have a significant effect on the process with *p* value greater than 0.05. As frequency increases, the cavitation yield drops because the cavitation bubbles become smaller and less energetic thus lowering the yield [19].

Next, the effect of interaction between the mass of leaves and extraction time is illustrated in Figure 2c. The yield of pandan extract shows a declining trend as the extraction time increase. This could be because, after 20 min, the extraction process has entered the second stage, which is the slow extraction process which causes the decrease in the yield of the extract. The same trend is also observed for the effect of the mass of leaves. Because the solid to solvent ratio is reducing, the solvent could be saturated, causing incomplete extraction and the efficiency of the process to drop.

On the other hand, the 3D surface of the yield of pandan extract against amplitude and concentration of solvent is displayed in Figure 2d. From the figure, it can be observed that operating the process at a higher solvent concentration and amplitude resulted in a higher yield of pandan extract. Though it seems favourable, it should be noted that amplitude has the least impact on the extraction process. This explains the insignificant rise in the yield of pandan extract (from 48% to 58.4%) as the amplitude increases. This can be supported by a study carried out to extract Pistachi Khinjuk Hull oil using an ultrasonic extraction process [20]. Ultrasonic amplitude and pre-treatment time had a significant effect on the oil yield. The highest oil yields at the three amplitude levels tested were 24.4, 29.8 and 37.8% for 0, 25 and 50%, respectively. Although the increase in pretreatment time increased oil yield, pre-treatment time at 30 and 45 min did not significantly differ for both the 20 and 50% amplitudes.

Finally, Figure 2e demonstrates the 3D yield of pandan extract against the mass of pandan leaves and amplitude. The figure shows the yield fluctuation as the system operates at a higher amplitude and mass of leaves. However, the yield changes when the amplitude increase is quite small (69% to 64.5%), whereas the reduction in yield for the mass of leaves is quite significant. A higher amplitude will increase agitation while decreasing cavitation. Higher amplitudes, on the other hand, can be chosen based on sample properties; for higher viscosity solvents, as it is best to optimize the amplitude/power level to achieve the desired agitation and cavitation for various sonochemical reactions and extraction efficiencies. To avoid undesirable degradation of the extracted compound, optimal amplitude/intensity conditions must be chosen [21].

The coefficient of variance (CV), the ratio of the standard error of estimate to the mean value of the observed response, was a measure of the reproducibility of the model. CV was reproducible when it is not greater than 10%. The value of CV for the response of concentration of 2AP is greater which suggests that it is a reproducible error of 17.64%.

Figure 3a–f shows the 3D response surface plots representing the effect of response variables on the concentration of 2AP obtained from pandan extract. From Table 3, it can be deduced that the mass of pandan leaves gives the most significant impact on the process since the F-value (31.97) is the highest compared to the other variables, while sonicator amplitude gives the least effect considering the F-value is the lowest which is 1.42. The dependency on extraction time and the amplitude can be evaluated to be minimal. The associated interaction parameters A (extraction time) and D (mass of pandan leaves) can be analysed to be the highest, while the lowest interaction between parameters is A (extraction time) and C (sonicator amplitude).

Figure 3a shows the surface plot of concentration of 2AP as a function of extraction time and mass of pandan leaves. As can be seen from the figure, the concentration of 2AP decreases gradually from 4.7 ppm to 2.0 ppm as the extraction time increase. However, after 35.15 min of extraction, the concentration of 2AP starts to increase again. Theoretically, a longer extraction time improves extraction yield. A longer extraction time gives the ultrasound wave more time to disrupt the cell walls releasing the cell content. However, as the extraction time is prolonged, the contact area would be decreased on the inner cell walls due to the increasing distance. A too prolonged extraction time may also damage the extracted compounds [16]. Meanwhile, a declining pattern can be seen for the concentration of 2AP as the mass of pandan leaves increases. This is because the solid-to-solvent ratio decrease. A higher solid-to-liquid ratio ensures homogeneous mixing and allows solvent penetration into deep interior portions. This suggests a higher concentration gradient between the inside plant cells and the solvent, which improves 2AP movement from solid to solvent. At a lower ratio, the rise of 2AP concentration is slow due to the limited amount of solvent and lack of deep penetration into the plant cell walls to dissolve 2AP [22].

3D surface plot of concentration of 2AP against the effect of concentration of solvent and extraction time is illustrated in Figure 3b. The concentration of solvent is the second most significant parameter with a *p* value of 0.0054, according to Table 2, while extraction time is the most important parameter with *p* value of 0.008 along with the mass of pandan leaves. Hence, the interaction between the concentration of solvent and extraction time becomes significant as the *p* value is 0.0002. From the figure, the increment of concentration of 2AP can be seen as the concentration of solvent used and extraction time applied during the extraction process increases. A similar result was reported in a study to extract antioxidant compounds from *Moringa oleifera* leaf [23]. The reason for the slow rise in the concentration of 2AP when the extraction time is prolonged is because the extraction is in the second stage where the external diffusion of soluble constituents through the porous structure of the residual solids and its transfer from the solution in contact with the particles to the bulk of the solution is happening. Ultrasonic waves could shatter cell walls, resulting in a larger contact area between solvent and substance and more oil on the surface. However, as the distance between the cells increases, the effect on the inner cell walls becomes weaker. As a result, ultrasonic waves have the biggest influence on mass transfer rate during the solvent penetration stage [24]. A similar trend also was discovered in a study to extract antioxidants from the dry peels of pomegranate marc by applying ultrasonic irradiation in continuous and pulsed modes [25].

Figure 3c portrayed the 3D surface plot of the effect of amplitude and extraction time on the concentration of 2AP. From the figure, it can be seen that the amplitude of the process does not have a significant impact on the concentration of 2AP since the changes in concentration is too small as the amplitude increase. However, it cannot be ignored that the increase in the concentration of 2AP could be attributed to better cavitation and mechanical effects of ultrasound, which increased the contact surface area between solid and liquid surfaces and resulted in more solvent penetration into the plant matrix. Because the resonant bubble size is related to the amplitude of the ultrasonic wave, cavitation bubble collapse becomes more extreme as the amplitude of the ultrasonic wave increases [26]. Because the temperature and pressure inside the bubbles are very high and the bubbles collapse in a very short time, a violent shock wave and a high-speed jet are generated, which will enhance solvent penetration into the cell tissues and accelerate intracellular product release into the solvent by disrupting the cell walls. Furthermore, the severe shock wave and high-speed jet may have caused the molecules to mix more thoroughly, increasing the mass transfer rate [24]. Meanwhile, when a longer extraction time is applied, the concentration of 2AP appears to be declining.

Next, the effect of interaction between the mass of leaves and amplitude towards the concentration of 2AP is displayed in Figure 3d. When amplitude increases during the process, the concentration of 2AP also increases but in a small amount. As for the mass of the leaves, the concentration of 2AP shows a declining trend as the mass of leaves used increases. This is due to the decrease in the solid to solvent ratio during the process. The viscosity of the solution is high at a low solid to solvent ratio, which complicates the cavitation effect because the negative pressure in the rarefaction cycle must overcome a higher cohesive force in the high viscous solution. The viscosity and concentration of the extraction medium drop as the solid to solvent ratio increases, resulting in a stronger cavitation impact. The greater the concentration difference between the solute and the solvent, the greater the diffusivity and dissolution of the solute in the solvent during the extraction process. At a high solid to solvent ratio, the ultrasonic intensity imposed on the by-product matrix is greater, resulting in more fragmentation, erosion, and pore formation, hence enhancing the mass transfer of 2AP. The increased contact area between the substance and the solvent may also boost the output of 2AP [27,28,29,30,31,32,33,34,35,36,37,38].

Other the other hand, the 3D surface plot of concentration of 2AP as a function of amplitude and concentration of solvent is shown in Figure 3e. As can be seen from the figure, as the concentration of solvent increases, the concentration of 2AP also shows an increment. Meanwhile, the concentration of 2AP shows a declining trend when the amplitude is within the range of 25% to 33% however when the amplitude rises above 33%, the concentration of 2AP starts to increase.

Finally, the effect of interaction between the concentration of solvent and mass of leaves towards the concentration of 2AP is illustrated in Figure 3f. The concentration of 2AP shows an inclining trend as the concentration of solvent elevates. When the mass of leaves increases from 7.5 g to 9.4 g, the concentration of 2AP appears to be declining. However, the concentration of 2AP begins to rise when the mass of leaves is above 10.3 g.

Sonication enhances the mass movement of bioactive chemicals through the combined influence of three processes, namely cavitation, mechanical agitation, and thermology [15]. Ultrasonic waves create compression and expansion in the medium during sonication, referred to as cavitation. The term “thermology” refers to raising the temperature at various locations on the plant matrix due to cyclic wave compression and expansion. Increased diffusivity and softening of vegetal surfaces are favoured by in situ and controlled temperature elevation, both advantageous for increased mass transport. Thus, cavitation can be regarded as the underlying phenomenon favouring agitation and thermology.

The solvent concentration is also an essential factor that affects the extraction process. A higher concentration of solvent will result in higher viscosity. The initiation of cavitation in a liquid requires that the negative pressure during the rarefaction cycle overcome the cohesive forces between molecules composing the liquid. A rise in viscosity or surface tension increases these molecular interactions, raising the cavitation threshold. The formation of a cavitation becomes more difficult when the solvent viscosity and surface tension are higher. Ultrasonic energy produced to aid the cavitation process will also increase [17]. In this manner, the amplitude should be increased when working with high viscosity samples. As the sample’s viscosity increases, the resistance of the sample to the movement of the ultrasonic device is also increased. Therefore, the amplitude of the system should be increased to achieve a better yield [18].

The solute-to-solvent ratio is one of the most critical factors during the mass transfer because a larger solvent volume helps accelerate the diffusion process [19]. An increase in phenolic compounds’ concentration is observed as the solute/solvent ratio increases [20]. A ratio of 1:40 g/mL solute/solvent is ideal for providing the amount of solvent required to enter the cells, thereby improving the phenolic compounds’ permeation [21]. However, it should be noted that ultrasonic extraction can generate more soluble compounds. Therefore, using high amounts of solvent could lead to liquid saturation in the extraction system. Furthermore, high amounts of solvents mean an increased cost for subsequent operations, such as the concentration and filtration of the extracts obtained and an increase in the amount of waste generated [22].

This study applies a CCD-based response surface methodology to obtain valuable insights and optimize the process parameters. While RSM may not achieve substantial gains in process parametric values, it is a very useful technique for the chosen system because the methodology implies evaluating the linearity and non-linearity of specific response variables in terms of process variables, which cannot be used deduced using trial and error approaches. As a result, parametric interactions and the identification of specific parameters and their interactions on the selected response variables can be understood. It is worth noting that the CCD-based RSM design required fewer experimental runs to accomplish an optimal set of process parameters.

### 3.6. GC-MS Analysis Result

Table 7 shows the compounds present in pandan extract at optimum conditions during GCMS analysis. The peak of 2AP is detected at a peak of 8.367 with 2.044% of peak area data. Other components that are identified include other aromatic components, alcohols, carboxylic acid, ketones, phenols, aldehydes, terpenes and pyridines which is also supported by other research [39]. This is supported by a previous study of extracting 2AP by supercritical carbon dioxide and ethanol extraction [12]. It was also recorded that there were 19 main components detected including three aldehydes, four alcohols, two ketones, two carboxylic acids, one hydrocarbon, one acyclic terpene, one furanone, three pyridines, one furan and one pyrroline. While 2AP owns 17.65% of the total volatiles, based on peak area data, other major components, such as cis-3-hexenol, trans-2-hexenal, 2-methyl-2-pentenal, 2,4-heptadienal, and 3-methyl-2(5H)-furanone amounted to 0.82%, 0.13%, 0.19%, 6.45% and 1.52% of the total volatiles, respectively during supercritical carbon dioxide extraction [12].

## 4. Conclusions

This study proved that UAE of pandan leaves facilitated excellent extraction profiles of 2AP due to sonication. Extraction time and solvent consumption in the conventional methods can be reduced by applying UAE method. It also proved that the inexpensive and non-toxic ethanol could be used as an alternative solvent for the optimal extraction of 2AP from pandan leaves. The optimal control variables of ultrasonic extraction of 2AP are at a 20 min extraction time, 60% of solvent concentration, amplitude at 25% and 12.5 g of pandan leaves, which produced 60.51% of the extract yield and 1.43 ppm of 2AP. RSM-based design and analysis provided essential insights as non-linear and quadratic profiles existed for several measured responses with variations in control variable values. As a result, RSM enabled a greater comprehension of the combined effect of the various control variables on the response variables. In conclusion, the study is highly promising in furthering the use of ultrasound to extract pandan leaves to produce natural functional foods. The finding of this study can be used to enhance the rice flavour as well as to be used as one of the flavour enhancers for cooking or baking.

## Figures and Tables

**Figure 1 molecules-27-04906-f001:**
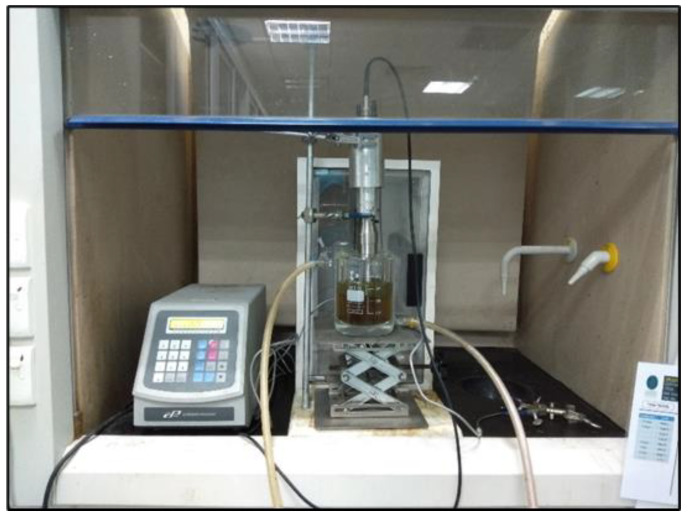
Ultrasonic probe system setup.

**Figure 2 molecules-27-04906-f002:**
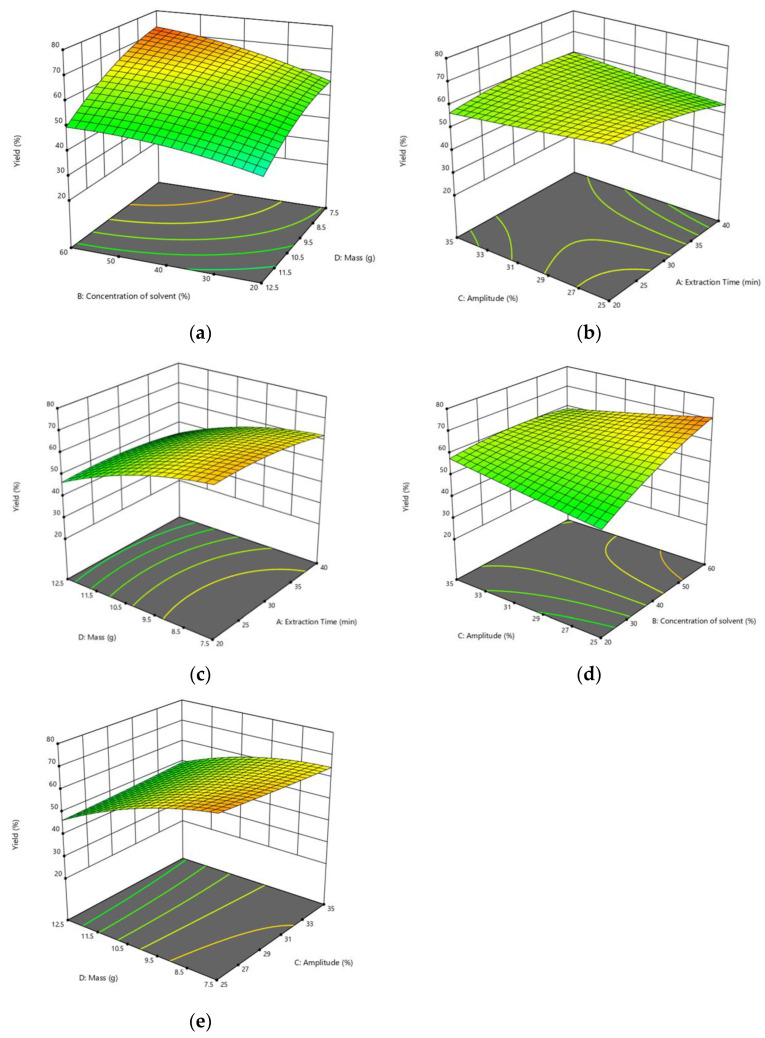
3D response surface plots representing the effect of response variables towards the yield of pandan extract; (**a**) Effect of concentration of solvent and mass of leaves towards yield of extract; (**b**) Effect of sonication amplitude and extraction time towards yield of extract; (**c**) Effect of mass of leaves and extraction time towards yield of extract; (**d**) Effect of sonication amplitude and concentraction of solvent towards yield of extract; (**e**) Effect of mass of leaves and sonication amplitude towards yield of extract.

**Figure 3 molecules-27-04906-f003:**
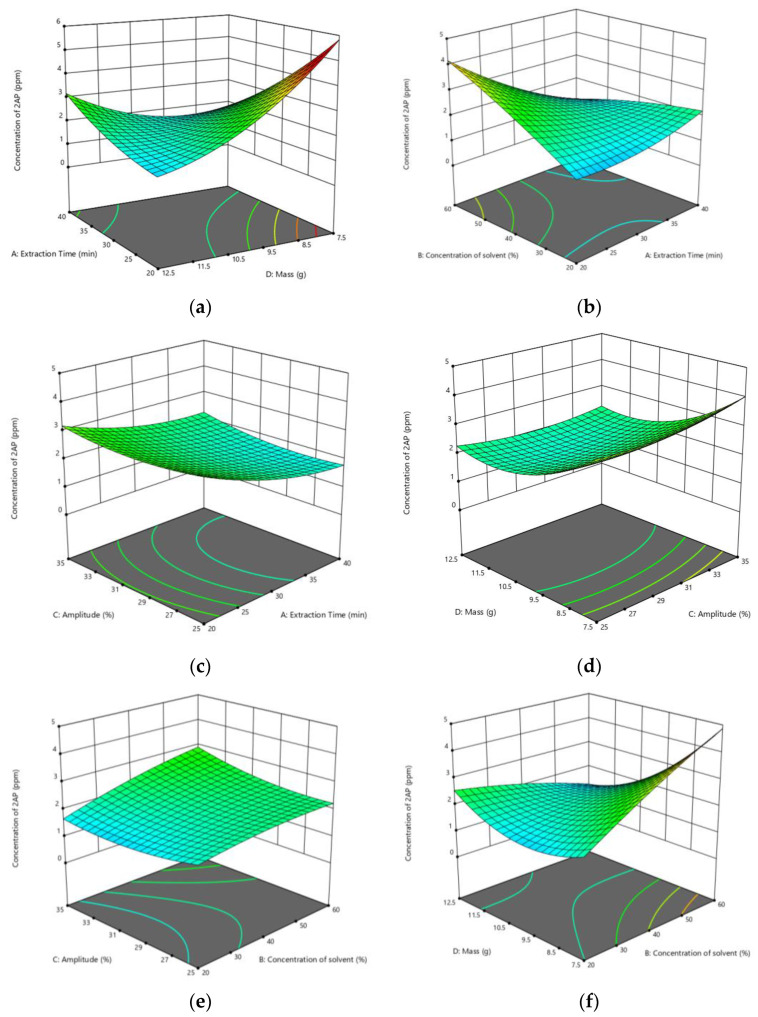
3D response surface plots representing the effect of response variables on the concentration of 2AP obtained from pandan extract; (**a**) Effect of mass of leaves and extraction time towards concentration of 2AP; (**b**) Effect of concentration of solvent and extraction time towards the concentration of 2AP; (**c**) Effect of sonication amplitude and extraction time towards concentration of 2AP; (**d**) Effect of mass of leaves and sonication amplitude towards concentration of 2AP; (**e**) Effect of sonication amplitude towards concentration of 2AP; (**f**) Effect of mass of leaves and concentration of solvent towards concentration of 2AP.

**Table 1 molecules-27-04906-t001:** Specification of ultrasonic processor.

Item	Specifications
Brand	Cole-Parmer
Frequency (Hz)	Up to 60
Range of time	1 s to 10 h
Power (W)	750
Temperature (°C)	1–100
Sample volume	250 µL to 19 L/h
Dimension	191 × 216 × 343 mm

**Table 2 molecules-27-04906-t002:** Table of varied parameters and concentration of 2AP obtained.

Std No	A: Extraction Time (Min)	B: Concentration of Solvent (%)	C: Amplitude (%)	D: Mass (g)	Yield of Extract (%)	Concentration of 2AP (ppm)
1	40	60	35	12.5	49.68	1.578
2	20	20	35	7.5	60.00	2.836
3	20	60	25	12.5	60.48	1.434
4	40	60	25	7.5	77.20	1.911
5	40	20	35	7.5	61.47	1.608
6	40	20	35	7.5	58.93	3.002
7	40	60	25	12.5	50.48	1.670
8	40	20	25	12.5	63.36	4.873
9	20	20	25	12.5	45.04	1.571
10	30	40	30	10	61.00	2.072
11	20	60	35	12.5	38.16	1.857
12	20	20	35	12.5	46.96	0.591
13	20	20	35	12.5	65.60	1.043
14	40	20	35	12.5	52.64	4.541
15	20	20	25	7.5	61.33	3.725
16	40	60	35	7.5	62.13	2.691
17	40	20	25	7.5	55.73	1.133
18	30	80	30	10	65.90	2.490
19	30	40	40	10	63.50	3.833
20	30	40	20	10	53.50	2.497
21	30	0	30	10	31.00	0.955
22	30	40	40	10	61.20	3.086
23	50	40	30	10	45.10	2.043
24	30	40	30	15	25.40	3.629
25	30	40	30	10	61.00	2.072
26	10	40	30	10	53.00	4.715

**Table 3 molecules-27-04906-t003:** RSM best for model fitness parameters (F-value and *p*-value).

Components	Yield of Extract		Concentration of 2AP	
	F-Value	*p*-Value	F-Value	*p*-Value
Model	6.47	0.0042	12.10	0.0014
	Modified	Modified	Quadratic	Quadratic
A	1.79	0.2141	31.41	0.0008
B	23.11	0.0010	15.71	0.0054
C	0.1936	0.6703	1.60	0.2463
D	55.66	<0.0001	31.97	0.0008
AB	-	-	50.87	0.0002
AC	4.52	0.0624	0.6563	0.4445
AD	0.2298	0.6431	113.71	<0.0001
BC	11.47	0.0080	3.67	0.0971
BD	1.60	0.2380	61.09	0.0001
CD	1.64	0.2329	1.25	0.3000
A^2^	2.41	0.1552	7.77	0.0270
B^2^	2.76	0.1309	1.58	0.2485
C^2^	0.0344	0.8570	5.13	0.0579
D^2^	6.81	0.0283	27.75	0.0012
R-Squared	0.9034		0.9603	
Adequate Precision	11.5596		11.5921	

**Table 4 molecules-27-04906-t004:** Optimum conditions of ultrasonic extraction of yield of extract and 2AP from pandan leaves.

Parameters	Extraction Time (Min)	Concentration of Solvent (%)	Amplitude (%)	Mass of Leaves (g)
Value	20	60	25	12.5

**Table 5 molecules-27-04906-t005:** Replicate runs at the optimum condition of yield of pandan extract from ultrasonic extraction process.

Replicate Runs	Run 1	Run 2	Run 3	Average Value (%)	Predicted Value (%)	Error (%)
Yield (%)	60.48	60.53	60.51	60.51	56.38	7.32
Concentration of 2AP (ppm)	1.43	1.47	1.39	1.43	1.77	19.10

**Table 6 molecules-27-04906-t006:** Previous extraction methods used to extract 2AP.

Extraction Method	Solvent Used	Extraction Time	Concentration of 2AP Obtained	Ref.
Supercritical carbon dioxide extraction	Liquid carbon dioxide	2 h	0.16 to 0.19 ppm	[19]
Supercritical carbon dioxide extraction	Liquid carbon dioxide	2 h	0.72 ppm	[12]
Hexane extraction	Hexane	12 h	0.52 ppm	[19]

**Table 7 molecules-27-04906-t007:** Compounds identified in the extract.

Peak	Compounds
2.739	Acetic acid
2.874	O-Methylisourea
	2-propanone
3.263	Methyl glyoxal
	1,4-Butanediamine
	Propanoic acid
4.480	2(5*H*)-Furanone
	3-methyl-
	*N*-Methyl methacrylamide
	4-Methyl-2-hexene,c&t
5.618	4*H*-Pyran-4-one
5.88	Catechol
6.012	Benzofuran
	2,3-dihydro-Benzeneacetaldehyde
	Benzene
6.647	2,5-Methano-2*H*-furo [3,2-b]pyran
	2,4,6-Trihydroxytoluene
	4(1*H*)-Pyrimidinone
	-amino-6-(methylamino)-
8.367	2-acetyl-1-pyrroline
8.419	1,3,5-tri-tert-butyl-Propanamide
8.758	4-((1*E*)-3-Hydroxy-1-propenyl)-2-methoxyphenol
	1,2,4-Cyclopentanetrione
8.966	6-Hydroxy-4,4,7a-trimethyl-5,6,7,7a-tetrahydrobenzofuran-2(4*H*)-one
	[5-(Hydroxymethyl)-3-(2*H*-1,2,4-triazol-3-yl)-1,2,3-triazol-4-yl]methanol
	1-(2-Hydroxycyclohexyloxy)-1*H*-pyridin-2-one
9.033	3-Methylamino-1-.beta.-d-ribofuran
	osylpyrazolo [3,4-d]pyrimidin-4-one
	Benzyl alcohol
	ethanone
9.198	Bicyclo [2.2.2]octa-2,5-diene, 1,2,3,6-tetramethyl
	5-Hydroxy-3-methyl-1-indanone
9.384	Pyrrolidine
	2-hexyl-1-methyl-Pyrrolidine,
	*N*-(4-methyl-4-pentenyl)-2-Cyclohexylpiperidine
9.622	*n*-Hexadecanoic acid
10.07	4′-Hydroxyphenazopyridine
	Benzoic acid
	4-(4-propylcyclohexyl)-, 4-cyano-3-fluorophenyl ester
	Benzimidazole-5-amine
10.235	Phytol
10.336	9,12,15-Octadecatrienoic acid
10.788	1,4-Dioxaspiro [4.5]decan-8-ol
	2(3*H*)-Furanone
	dihydro-4-methyl-5-pentyl-2-Piperidinone
11.98	1*H*-Indole
	*N*-ethyl-5,6-dimethoxy-3-methyl-2-(4′-methoxyphenyl)-6-Methyl-2-phenyl-7-(2,4-dimethylphenylmethyl)indolizine
	butane
13.401	(5*S*,8*R*,8a*S*)-8-(But-3-en-1-yl)-5-((*Z*)-pent-2-en-4-yn-1-yl)octahydroindolizine
	(5*R*,8*R*,8a*S*)-8-(But-3-en-1-yl)-5-(pent-4-yn-1yl)octahydroindolizine
	Isoquinolin-6-ol
	1-[4-hydroxybenzyl]-1,2,3,4-tetrahydro-7-methoxy

## Data Availability

Not applicable.

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
