# Peer review of "Ultrasonic Extraction of 2-Acetyl-1-Pyrroline (2AP) from Pandanus amaryllifolius Roxb. Using Ethanol as Solvent"

_molecules, 2022, doi:10.3390/molecules27154906_

Round 1
Reviewer 1 Report
From our point of view, we believe that the article has been adequately revised and all the changes that we had suggested have been discussed and introduced.
However, there is some typographical error or improvement of the images in Figures 2, 3, 4 that we ask you to consider.
Next we expose the lines where we observe small errors:
line 273. Spacing should be given
lines 435 to 438. Check if in line 438, if it should say reproducible an error of 17.64%
Line 579. Change figure 3 to figure 4.
We also suggest the following improvements:
Improve the quality of figures 2, 3 and 4 allowing their content and quality to be read and displayed, since they are somewhat blurred.
Try to integrate Tables 4 to 8 in a single Table, since they are excessive.
Author Response
Point 1: However, there is some typographical error or improvement of the images in Figures 2, 3, 4 that we ask you to consider.
Point 2: line 273. Spacing should be given.
Response 2: Spacing has been added.
Point 3: lines 435 to 438. Check if in line 438, if it should say reproducible an error of 17.64%.
Response 3: Sentence has been fixed.
Point 4: Line 579. Change figure 3 to figure 4.
Response 4: Figure 3 is changed to Figure 4.
Point 5: Improve the quality of figures 2, 3 and 4 allowing their content and quality to be read and displayed, since they are somewhat blurred.
Response 5: All figures inserted in manuscript are already in highest resolution.
Point 6: Try to integrate Tables 4 to 8 in a single Table, since they are excessive.
Response 6: Table 4 and 6 can be integrated into a single table. Table 5 and 7 also integrated into a single table.
Reviewer 2 Report
The aim of this paper was to optimize the ultrasonic extraction of 2-acetyl-1-pyrroline from pandan leaves using ethanol as a solvent. The manuscript is interesting and fits within the scope of the journal. The authors have improved the manuscript but there are still problems that need to be solved.
Please include more information about plant material (botanical identification by…, voucher ID)
Please include a separate subchapter for statistical analysis.
Results from GC-MS analysis are not included. Please complete the manuscript. Please include all information about compounds identified by GC-MS. It is normal to be able to make a complete comparison between the extract obtained by you and what was previously published (not only for 2AP).
Author Response
Point 1: Please include more information about plant material (botanical identification by…, voucher ID)
Response 1: Voucher ID is not attached to the plant during the research hence the absence of voucher ID.
Point 2: Please include a separate subchapter for statistical analysis.
Response 2: Subchapter for statistical analysis has been included in reflected in page 6.
Point 3: Results from GC-MS analysis are not included. Please complete the manuscript. Please include all information about compounds identified by GC-MS. It is normal to be able to make a complete comparison between the extract obtained by you and what was previously published (not only for 2AP).
Response 3: Comparison between extract obtained in other extraction methods has been added and reflected in page 16.
Reviewer 3 Report
The comments are as follows:
- Please, improve the abstract importing more key-findings.
- The introduction is too generalized, well-known and should be rewritten.
- Section Preparation of Pandan Leaves: It is not necessary to write every default step of the procedure. It should be revised and filled with only relevant details.
- Figure 4 needs a better quality.
- It is necessary to revise the text to correct a lot of misprints.
- The importance of this study and the practical applications of the findings has to be expressed more.
Author Response
Point 1: Please, improve the abstract importing more key-findings.
Point 2: The introduction is too generalized, well-known and should be rewritten.
Point 3: Section Preparation of Pandan Leaves: It is not necessary to write every default step of the procedure. It should be revised and filled with only relevant details.
Response 3: Preparation of pandan leaves section has been simplified. Please refer to page 3.
Point 4: Figure 4 needs a better quality.
Response 4: All the figures included in the manuscript are already in highest resolution.
Point 5: It is necessary to revise the text to correct a lot of misprints.
Response 5: Text are revised.
Point 6: The importance of this study and the practical applications of the findings has to be expressed more.
Round 2
Reviewer 2 Report
The authors improved the manuscript according to the reviewers' specifications. However, the results of GC-MS could be fully presented. I suggest removing the chromatogram and including a table with the identified and quantified compounds.
Author Response
Point 1: The authors improved the manuscript according to the reviewers' specifications. However, the results of GC-MS could be fully presented. I suggest removing the chromatogram and including a table with the identified and quantified compounds.
Response 1: Table with identified compounds has been added. See Table 7 as reference.
Reviewer 3 Report
- Please, improve the abstract importing more key-findings.
- It is necessary to revise the text to correct a lot of misprints.
- The importance of this study and the practical applications of the findings has to be expressed more.
Author Response
Point 1: Please, improve the abstract importing more key-findings.
Response 1: Abstract has been improved.
Point 2: It is necessary to revise the text to correct a lot of misprints.
Response 2: Text has been revised.
Point 3: The importance of this study and the practical applications of the findings has to be expressed more.
Response 3: Practical applications are mentioned at the end of the conclusion. "The finding of this study can be used to enhance the rice flavor as well as to be used as one of the flavor enhancer for cooking or baking."
This manuscript is a resubmission of an earlier submission. The following is a list of the peer review reports and author responses from that submission.
Round 1
Reviewer 1 Report
This study aims to find the optimal ultrasonic extraction conditions to obtain a maximum concentration of 2AP from pandan extract. From my point of view, the work presents the following deficiencies and the work cannot be accepted if the suggested improvements are not made:
(i) It does not detail the intensity and power of the ultrasound equipment and does not define the units of measurement of the amplitude and we do not understand what is being talked about. To speak of amplitude, the power and intensity of the US equipment must be specified. For example, in Table 1 the units of the different variables are not established.
(ii) It remains to incorporate in this study, how it varies in the extract obtained, the total content of polyphenols and how it varies depending on the different variables. The work is very poor and it is necessary to incorporate how it varies with intensity and potency and time, not only the polyphenols, but also the rest of the chemical substances contained in the leaf extracts, apart from the 2AP study, and to discuss the results of this study with others. jobs. For example, in lines 314-315 he cites an article that talks about the increase in polyphenol content when the solute-solvent ratio increases, but in this study the total polyphenol content is not even calculated and how it varies in the different tests. These trials should be included in material and methods and discussed to improve this article.
(iii) Lines 302-312, we talk about the concentration of the solvent and the viscosity. The wording should be rewritten and improved, according to the results obtained and the analysis of polyphenols (which remains to be done) and discussed and compared with other studies already carried out. In reference [2] [No. Ngadi and N. Y. Yahya, "Extraction of 2-Acetyl-1-Pyrroline (2AP) in Pandan Leaves (Pandanus Amaryllifolius Roxb.) Via Solvent Extraction Method: Effect of Solvent," Jurnal Teknologi 2014], a study on this topic is already offered and for this study to be accepted, at a minimum the content of total polyphenols should be included.
(iv) It is necessary to apply higher resolution and quality of the figures made and improve the discussion of results by incorporating other studies and how other chemical species and/or total polyphenols vary.
Reviewer 2 Report
The aim of this paper was to optimize the ultrasonic extraction of 2-acetyl-1-pyrroline from pandan leaves using ethanol as a solvent. The manuscript is interesting and fits within the scope of the journal. I recommend completing and resubmitting the manuscript:
Major revision:
Results and discussion for GC-MS analysis are not included. Please complete the manuscript.
Extensive editing of English language and style required.
Minor revision:
Title:
Please include in the title the scientific name of the plant: Pandanus amaryllifolius
Abstract:
Please include more information about the results and a general conclusion.
Introduction:
Please include the complete scientific name of a plant: Pandanus amaryllifolius.
Please highlight the degree of novelty and originality of the work.
Materials and Methods:
Please include more information about plant material (scientific name, botanical identification by…, voucher ID, collection date…)
Please include technical information about the ultrasonic bath.
How did you keep your temperature constant?
Please include a separate subchapter for statistical analysis.
Results and Discussion
- Please improve the quality of the figures.
- Results from GC-MS analysis are not included. Please complete the manuscript.
- Include in the text potential research directions. What are the future applications? What are the next research directions?
Reviewer 3 Report
- The introduction section (lines 67-89) should be resumed. It is too generalized, well-known and should be rewritten highlighting the aims of the research. It is better to comment on the influence of extraction parameters in the Results and discussion section.
- Section Preparation of Pandan leaves has to be rewritten. The text form line 95 to 100 is not necessary. Moisture content of Pandan leaves is missing.
- Section Ultrasonic Extractions of 2AP From Pandan Leaves: It is not necessary to write every default step of the procedure. More details about ultrasonic probe are needed.
- A comparison of UAE with other extraction techniques is needed. It is recommended to use solid-liquid extraction as a reference for the optimization study of UAE.
- Section Optimality of independent variables needs more specific discussion.
- There is a lack of discussion and comparison of obtained results with scientific literature.
- The importance of this study and the practical applications of the findings has to be expressed more.
- It is necessary to revise the text to correct some misprints.